# Relationship between Renal Function, Fibrin Clot Properties and Lipoproteins in Anticoagulated Patients with Atrial Fibrillation

**DOI:** 10.3390/biomedicines10092270

**Published:** 2022-09-13

**Authors:** Wern Yew Ding, Ian G. Davies, Dhiraj Gupta, Gregory Y. H. Lip

**Affiliations:** 1Liverpool Centre for Cardiovascular Science, University of Liverpool and Liverpool Heart & Chest Hospital, Liverpool L14 3PE, UK; 2Research Institute of Sport and Exercise Science, Liverpool John Moores University, Liverpool L3 5UX, UK; 3Department of Clinical Medicine, Aalborg University, 9220 Aalborg, Denmark

**Keywords:** chronic kidney disease, turbidimetry, low-density lipoprotein, small dense low-density lipoprotein, oxidised low-density lipoprotein

## Abstract

**Background**: Mechanisms by which chronic kidney disease (CKD) influences fibrin clot properties in atrial fibrillation (AF) remain ill-defined. We aimed to investigate the effects of AF and CKD on fibrin clot properties and lipoproteins, and determine the relationship between these factors. **Methods**: Prospective cross-sectional study of patients recruited from cardiology services in Liverpool between September 2019 and October 2021. Primary groups consisted of anticoagulated AF patients with and without CKD in a 1:1 ratio. Control group comprised anticoagulated patients without AF or CKD. Fibrin clot properties were analysed using turbidity and permeation assays. Detailed lipoprotein characteristics, including total cholesterol, low-density lipoprotein cholesterol (LDL-C), small dense LDL and oxidised LDL, were measured. **Results**: Fifty-six anticoagulated patients were enrolled (median age 72.5; 34% female); 46 with AF (23 with CKD and 23 without CKD) and 10 controls. AF was associated with changes in three indices of fibrin clot properties using PTT (T_lag_ 314 vs. 358 s, *p* = 0.047; Abs_peak_ 0.153 vs. 0.111 units, *p* = 0.031; T_lysis50%_ 884 vs. 280 s, *p* = 0.047) and thrombin reagents (T_lag_ 170 vs. 132 s, *p* = 0.031; T_max_ 590 vs. 462 s, *p* = 0.047; T_peak50%_ 406 vs. 220 s, *p* = 0.005) while the concomitant presence of CKD led to changes in fibrin clot properties using kaolin (T_lag_ 1072 vs. 1640 s, *p* = 0.003; T_max_ 1458 vs. 1962 s, *p* = 0.005; T_peak50%_ 1294 vs. 2046, *p* = 0.008) and PPP reagents (T_lag_ 566 vs. 748 s, *p* = 0.044). Neither of these conditions were associated with changes in fibrin clot permeability. Deteriorating eGFR was significantly correlated to the speed of clot formation, and CKD was independently associated with unfavourable clot properties (T_lag_ −778, *p* = 0.002; T_max_ −867, *p* = 0.004; T_peak50%_ −853, *p* = 0.004 with kaolin reagent). AF alone was not associated with changes in lipoprotein distribution while AF patients with CKD had lower total cholesterol, LDL-C and small dense LDL due to the presence of other risk factors. No significant relationship was observed between fibrin clot properties and lipoprotein distribution. **Conclusions**: There are important changes that occur in fibrin clot properties with AF and CKD that may account for the increased risk of thromboembolic complications. However, these changes in fibrin clot properties were not attributable to alterations in lipoprotein distribution.

## 1. Introduction

Atrial fibrillation (AF) is the most common sustained arrhythmia and the condition is associated with significant morbidity and mortality [1,2,3,4,5]. It is characterised by a prothrombotic state with the interplay of various underlying pathways [6]. Therefore, the management of patients with AF focuses on anticoagulation therapy to reduce the risk of thromboembolic complications. Nonetheless, the efficacy of anticoagulation agents in AF patients is attenuated with the presence of severe chronic kidney disease (CKD) [7], implying the involvement of other pathways among these patients. 

Changes in fibrin clot properties have been demonstrated in patients with AF and renal failure. Among patients with permanent AF, those who had suffered a previous thromboembolic event had longer clot lysis time [8]. Additionally, the CHA_2_DS_2_-VASc score correlated with clot lysis time, supporting the concept that impaired fibrinolysis reflects an increased risk of thromboembolic complications in AF [8]. We have previously demonstrated that AF patients with worsening CKD have accelerated fibrin clot formation and polymerisation that may explain the increased risk of thromboembolic events among these patients [9]. Additionally, worsening CKD was associated with formation of thicker fibrin clot structures that were more susceptible to fibrinolysis, thereby simultaneously increasing the risk of bleeding. A further study with scanning electron microscopy and nanostructure assessment of the fibrin clots in AF patients revealed increased density, fibre diameter and protofibril numbers, and decreased number of pores among patients with CKD stage 4 compared to those with CKD stage 1 [10]. That was the first study to demonstrate serial changes in fibrin clot structure in AF across worsening degrees of renal function [10].

Overall, the mechanisms by which CKD influences fibrin clot properties in AF remain to be determined. Several pathophysiological processes may be involved, and we previously discussed the contribution of lipoproteins to thrombosis in AF [11] but there is little research in this regard. The objective of this study was to explore the effects of AF and CKD on fibrin clot properties and lipoproteins, and determine the relationship between these factors.

## 2. Materials and Methods

### 2.1. Study Design and Population

A prospective cross-sectional study was performed with patients randomly recruited from cardiology services at Liverpool Heart and Chest Hospital NHS Foundation Trust and community anticoagulation clinics at the Liverpool and Sefton area, United Kingdom, between September 2019 and October 2021. The primary groups consisted of patients in a 1:1 ratio with known AF who were treated with oral anticoagulation (OAC) therapy with and without CKD. The control group comprised patients without AF or CKD who were treated with OAC therapy for other indications. Exclusion criteria were age below 18 years, severe mitral stenosis or presence of metallic prosthetic valve, active or recent malignancy (<6 months), active immunological disease, connective tissue disease, chronic liver disease, recent or chronic serious infection, chronic inflammatory disease, known haemophilia or thrombophilia, active bleeding, untreated hypothyroidism or hyperthyroidism, recent surgery (<3 months), familial lipid disorders, concurrent use of steroids and intake of dietary supplements known to influence lipids. The study protocol was approved by the North West—Liverpool Central Research Ethics Committee (Reference number: 19/NW/0355), HRA and Health and Care Research Wales. Written informed consent was obtained from the patients and the study was performed in accordance with the Declaration of Helsinki.

### 2.2. Data Collection and Definition

Data on demographics, medical conditions, medications, social history and examination findings were collected at baseline. Fasting venous blood sampling was collected for all patients. Routine blood tests, including albumin, clotting profile and renal function, were assessed by standard automated laboratory methods. AF was diagnosed by the treating physician in accordance with contemporary guidelines [12]. Paroxysmal AF was defined as AF that terminates spontaneously or with intervention within 7 days of onset, persistent AF as continuously sustained beyond 7 days including episodes terminated by cardioversion (drugs or electrical cardioversion) after ≥7 days and permanent AF where it is accepted by the patient and physician, and no further attempts to restore/maintain sinus rhythm were undertaken. Estimated glomerular filtration rate (eGFR) was determined using the Chronic Kidney Disease Epidemiology Collaboration equation [13]. For the purposes of this study, CKD was defined as eGFR less than 50 mL/min/1.73 m^2^.

Venous blood samples for fibrin clot analysis were taken into 3.2% citrate tubes and samples for lipoprotein analysis into ethylenediaminetetraacetic acid (EDTA) tubes. All the samples were centrifuged at 2000× *g* for 10 min at 22 °C. Following this, citrated plasma was further centrifuged at 10,000× *g* for 15 min at room temperature and passed through a 2.5 micron syringe filter to obtain platelet-poor plasma. Samples were stored at −80 °C until further analysis.

### 2.3. Clotting Analysis

The principles of turbidimetric clotting analysis using micro-plate assay have previously been published [14,15]. Briefly, citrated platelet-poor plasma was diluted with HBS buffer at pH 7.4 and the clotting reaction was initiated by addition of 5% bovine serum albumin, CaCl_2_ and a clotting reagent. Clotting reagents utilised were thrombin (Sigma-Aldrich, Gillingham, UK), kaolin (Sigma-Aldrich, UK), PPP reagent (Stago, France) and PTT automate (Stago, Asnières-sur-Seine, France) (Appendix A). Tissue plasminogen activator (Technoclone, Vienna, Austria) was added to fibrinolysis assays. Fibrin clot formation and lysis were analysed using a plate reader calibrated to measure an optical density of 340 nm for 1 h at 37 °C. Triplicate testing was performed for each sample. Indices on thrombogenesis and fibrinolysis were collected (Appendix A).

Fibrin permeation analysis measures the flow rate of buffer through a fibrin clot under constant pressure and provides information on the average pore size of fibrin clot (Darcy constant or Ks). This was performed as previously described [16]. Briefly, citrated platelet-poor plasma was mixed with HBS buffer at pH 7.4, thrombin and CaCl_2_. The mixture was transferred onto a μ-Slide (ibidi GmbH, Gräfelfing, Germany) and incubated in a humidity chamber for 1 h. Then, the flow rate of a reservoir of buffer through the fibrin clot was measured after an initial stabilisation period. The permeation constant was determined by Ks = (Q × L × n) / (T × A × dP), expressed in cm^2^, where Q = volume of liquid, *n* = viscosity, L = clot length, T = time, A = area and dP = pressure drop (D × G × H); D = density; G = gravity; H = liquid height. Duplicate testing was performed for each sample. The intra- and inter-assay coefficients of variation were 3.73 and 3.75%, respectively.

### 2.4. Lipoprotein Separation and Analysis

Lipoprotein separation was performed using the iodixanol gradient centrifugation method [17,18,19]. In brief, EDTA plasma was mixed with iodixanol and PBS, and centrifuged in a near-vertical rotor at 65,000 rpm for 3 h and 10 min at 16 °C. The generated lipoprotein fractions were subsequently fractionated into 20 different parts of equal volume using an Auto Densi-Flow machine (Labconco, Kansas City, MO, USA) and an FC 203 B fraction collector (Gilson, Middleton, WI, USA). High-precision measurement of the index of refraction for each lipoprotein fraction was performed using an Abbe refractometer. The lipoprotein density was calculated from the index of refraction and pooled low-density lipoprotein (LDL), large buoyant low-density lipoprotein (lbLDL) and small dense LDL (sdLDL) subclasses were prepared from the fractions based on density.

Lipoprotein characterisation for total cholesterol (TC), LDL-C and sdLDL-C concentrations was performed for whole plasma, total LDL, lbLDL and sdLDL subclasses using the corresponding kit for an RX Daytona Plus analyser (Randox, Manchester, UK). A commercially available enzyme-linked immunosorbent assay (ELISA) test was used to measure the OxLDL concentration (MyBioSource, San Diego, CA, USA). 

### 2.5. Statistical Analysis

Continuous variables were described with median and IQR, and tested for differences with a Mann–Whitney U test. Categorical variables were described as count and percentage, and tested for differences with a chi-square or Fisher’s exact test. To determine the effect size of CKD on turbidimetric clotting and lipoprotein analyses, and account for differences between baseline parameters between the groups, linear regression analysis was performed. The model was adjusted for parameters that were significantly different between the groups at baseline. Any association between fibrin clot properties and lipoprotein distribution was analysed with Pearson’s correlation coefficient. A 2-sided *p* value of less than 0.05 was considered statistically significant. All analyses were performed using SPSS version 27 (IBM Corp., Armonk, NY, USA).

## 3. Results

A total of 56 anticoagulated patients (median age 72.5 (IQR 66–80); 34% female) were prospectively recruited, 46 patients had known AF (CKD: *n* = 23; non-CKD, *n* = 23) and 10 patients were in the control group. Indications for anticoagulation in the control group included pulmonary embolism and/or deep venous thrombosis (*n* = 8), left ventricular thrombus (*n* = 1) and portal vein thrombosis (*n* = 1). There were no significant differences in baseline characteristics or routine biochemical parameters between AF patients without CKD and the control group (Table 1). Baseline characteristics of AF patients with and without CKD are shown in Table 2. Compared to non-CKD patients, AF patients with CKD were older (78 vs. 68 years, *p* = 0.012) with greater prevalence of comorbidities including diabetes mellitus (52.2 vs. 13.0%, *p* = 0.005) and heart failure (56.5 vs. 21.7%, *p* = 0.016). Furthermore, there was more use of statin therapy in patients with CKD (87.0 vs. 56.5%, *p* = 0.022). The choice of anticoagulation agent and use of antiplatelet therapy were comparable between the groups. AF patients with CKD had significantly lower haemoglobin levels (130 vs. 143 g/L, *p* = 0.017) and eGFR (32 vs. 72 mL/min/1.73 m^2^, *p* < 0.001) compared to the non-CKD group.

### 3.1. Fibrin Clot Properties in AF and CKD

The results of turbidimetric clotting analysis using kaolin, PPP, PTT and thrombin reagents in AF patients without CKD and the control group are shown in Table 3. Using PTT reagent, patients with AF had significantly shorter lag time (*p* = 0.047), higher peak absorbance (*p* = 0.031) and increased time to 50% clot lysis (*p* = 0.047), though no statistical differences were observed with time to maximum absorbance and time to 50% peak absorbance. Using thrombin reagent, patients with AF had significantly prolonged lag time (*p* = 0.031), time to maximum absorbance (*p* = 0.047) and time to 50% peak absorbance (*p* = 0.005), though no significant differences were observed with peak absorbance and time to 50% clot lysis. There were no significant differences between the groups with fibrin clot properties using kaolin and PPP reagents. Patients with AF had lower fibrin permeability compared to controls but this did not achieve statistical significance (*p* = 0.633). 

The results of turbidimetric clotting analysis using kaolin, PPP, PTT and thrombin reagents among AF patients with and without CKD are shown in Table 4. Using kaolin reagent, patients with CKD had significantly shorter lag time (*p* = 0.003), time to maximum absorbance (*p* = 0.005) and time to 50% peak absorbance (*p* = 0.008), though no statistical differences were observed with peak absorbance and time to 50% clot lysis. Using PPP reagent, patients with CKD had significantly shorter lag time (*p* = 0.044), though no statistical differences were observed with peak absorbance, time to maximum absorbance, time to 50% peak absorbance and time to 50% clot lysis. There were no significant differences in fibrin clot properties between the groups using PTT and thrombin reagents. Patients with CKD had lower fibrin permeability compared to non-CKD patients but this did not achieve statistical significance (*p* = 0.297).

### 3.2. Effects of eGFR on Fibrin Clot Properties

The results of correlation analysis between eGFR and fibrin clot properties in AF are presented in Appendix A. There were significant correlations between eGFR and lag time (0.450 (95% CI, 0.132 to 0.684), *p* = 0.008), time to maximum absorbance (0.385 (95% CI, 0.054 to 0.640), *p* = 0.025) and time to 50% peak absorbance (0.409 (95% CI, 0.102 to 0.644), *p* = 0.011) with kaolin reagent, time to maximum absorbance (0.327 (95% CI, 0.037 to 0.566), *p* = 0.028) with PPP reagent and time to maximum absorbance (−0.300 (95% CI, −0.543 to −0.011), *p* = 0.043) with thrombin reagent.

Using linear regression analysis, CKD was associated with a significant reduction in lag time (−589 (95% CI, −930 to −249) s), time to maximum absorbance (−596 (95% CI, −998 to −196) s) and time to 50% peak absorbance (−569 (95% CI, −980 to −158) s) with kaolin reagent (Appendix A). After adjusting for age, diabetes mellitus, heart failure, statin use and haemoglobin level, CKD was independently associated with a significant reduction in lag time (778 s, *p* = 0.002), time to maximum absorbance (867 s, *p* = 0.004) and time to 50% peak absorbance (853 s, *p* = 0.004) with kaolin reagent. No significant differences in fibrin clot properties were observed between the AF patients with and without CKD using PPP, PTT and thrombin reagents.

### 3.3. Lipoprotein Distribution

Cholesterol and OxLDL were measured in whole plasma; the total LDL fraction; and the two major subclasses of LDL, lbLDL and sdLDL. The results among AF patients without CKD and the control group are shown in Table 5. There were no significant differences between the groups in terms of lipoprotein cholesterol or OxLDL distribution for whole plasma, total LDL, lbLDL and sdLDL subclasses. 

Compared to non-CKD patients, AF patients with CKD had lower TC, LDL-C and sdLDL levels in whole plasma and total LDL, lbLDL and sdLDL subclasses (Table 6). OxLDL levels were comparable between AF patients with and without CKD.

### 3.4. Effects of eGFR on Lipoprotein Distribution

The results of correlation analysis between eGFR and lipoprotein distribution in AF are presented in Appendix A. There was significant positive correlation between lipoprotein distribution. Using linear regression analysis, CKD was associated with a significantly lower TC, LDL-C and sdLDL in whole plasma, reduced TC and LDL-C in the total LDL fraction and less TC, LDL-C and sdLDL in the lbLDL subclass (Appendix A). The effects of CKD on lipoprotein distribution in AF were attenuated after adjusting for age, diabetes mellitus, heart failure, statin use and haemoglobin levels.

### 3.5. Association between Fibrin Clot Properties and Lipoprotein Distribution

The results of correlation analysis between fibrin clot properties and lipoprotein distribution in whole plasma among patients with AF are presented in Table 7. There was significant positive correlation between sdLDL and lag time with PPP reagent (0.312 (95% CI, 0.020 to 0.555), *p* = 0.037), and OxLDL and lag time with PTT reagent (0.359 (95% CI, 0.077 to 0.588), *p* = 0.014). Otherwise, no significant relationship was observed between other fibrin clot properties and lipoprotein distribution.

## 4. Discussion

In this prospective study, we investigated the effects of AF and CKD on fibrin clot properties and lipoproteins among anticoagulated patients. First, AF and CKD were both associated with changes in fibrin clot properties. Second, there was a positive correlation between eGFR and lag time, time to maximum absorbance and time to 50% peak absorbance. Furthermore, CKD was an independent risk factor for unfavourable clot parameters, after adjustment for confounders. Third, there was no association between AF and lipoprotein distribution in terms of TC, LDL-C, sdLDL-C and OxLDL. Fourth, CKD was linked to lower TC, LDL-C and sdLDL-C. Moreover, there was a significant positive correlation between eGFR and lipoprotein levels including TC, LDL-C and sdLDL-C. However, the effects of CKD on lipoprotein distribution were driven by co-existing risk factors among these patients rather than CKD per se. Fifth, there were no strong correlation between fibrin clot properties and lipoprotein distribution among anticoagulated patients with AF.

The novelties of this study are underscored by the evaluation of the effects of several reagents used to initiate different pathways of the coagulation cascade and the investigation of lipoprotein distribution, including OxLDL, in the various LDL subclasses.

### 4.1. Changes in Fibrin Clot Properties with AF and CKD

AF and CKD are both characterised by a prothrombotic state. Nonetheless, few studies have evaluated the changes in fibrin clot properties in patients with AF and/or CKD. A study of sex differences in fibrin polymerisation and lysability of fibrin among VKA-treated patients with AF demonstrated that females had a higher rate of lateral aggregation and reduced rate of lysis compared to males [20]. These findings may account for the fact that female sex is a ‘risk modifier’ in AF [21,22]. A study of patients with AF who were commenced on VKA therapy showed increased fibrin permeability and reduced clot lysis time within 3 days of treatment and reaching a plateau after a week [23]. These modifications were related to lower vitamin K-dependent factor activity, protein C levels and INR values. In a separate study of patients with AF, it was demonstrated that there were reduced latency time and increased rate of clot formation and clot density in a concentration-dependent manner for NOACs, INR-dependent manner for phenprocoumon and anti-Xa level-dependent manner for LMWH [24]. Furthermore, clot lysis time was inversely correlated with NOAC concentration, INR and anti-Xa level, respective to anticoagulation agent. 

Among patients with coronary artery disease, the presence of CKD was associated with shorter latency time, increased clot density, reduced clot lysis time, decreased fibrin permeability and greater clot mass [25]. Additionally, reduced eGFR in non-anticoagulated patients with AF was associated with an enhanced prothrombotic state, including increased thrombin formation and impaired fibrinolysis [26].

Despite the findings above, the majority of studies employing turbidimetric clotting analysis were performed using thrombin reagent only and in VKA-treated patients, thereby limiting our understanding of the prothrombotic mechanisms that are implicated in AF and CKD. In this study, we investigated the effects of AF and CKD in patients who were treated with either VKA or NOAC therapy by performing turbidimetric clotting analysis for fibrin clot properties, initiated using various reagents that interact with different pathways in the coagulation cascade. It was found that AF and CKD were both associated with changes in fibrin clot properties; in particular, patients with AF had reduced latency time and increased clot density and clot lysis time while patients with CKD had reduced latency time and time to achieve maximum clot thickness. Of note, there were differences in fibrin clot properties depending on the initiating pathways. Furthermore, reduced eGFR was linked to decreased latency time and time to achieve maximum clot thickness. 

Overall, AF and CKD were independent risk factors for unfavourable clot parameters which may explain the excess risk of thromboembolism among patients with these conditions.

### 4.2. Impact of Unfavourable Fibrin Clot Properties on Long-Term Outcomes

Though several conditions have been shown to influence fibrin clot properties, the long-term implications of these changes are poorly described. Among patients with coronary artery disease and either prior myocardial infarction and/or diabetes mellitus, larger clot area under the curve, reflecting the balance between clot formation and lysis, was independently related to excess acute non-fatal myocardial infarction, ischaemic stroke and cardiovascular death [27]. Meanwhile, greater clot density was independently linked to higher risk of cardiovascular death and all-cause death. A substudy of patients with acute coronary syndrome from the PLATelet inhibition and patient Outcomes (PLATO) trial found a graded increase in the 1-year risk of cardiovascular death or myocardial infarction, and cardiovascular death alone with greater clot density and clot lysis time [28]. Neither of these clot parameters was linked to the risk of major bleeding. Further, a study of patients with diabetes mellitus from the PLATO cohort demonstrated similar results [29]. In patients on chronic haemodialysis, greater clot density was independently associated with increased all-cause death and cardiovascular death, after adjusting for confounders including age, diabetes status, sex and duration of dialysis [30]. These findings suggest that denser clot structures may partially explain the risk of poor outcomes in patients on haemodialysis. 

### 4.3. Effects of AF and CKD on Lipoprotein Distribution

Previous studies have suggested that lipoproteins, which have an integral role in the coagulation cascade, may be influenced by the presence of AF and/or CKD. A study of consecutive AF patients with eGFR equal to or more than 60 mL/min/1.73 m^2^ found that compared to matched controls in sinus rhythm, patients with AF had higher OxLDL levels, though no significant differences were observed in TC, HDL-C, LDL-C and triglycerides [31]. Overall, the presence of AF independently predicted greater OxLDL levels. Moreover, over a follow-up period of 4 years, each tertile increase in OxLDL was associated with a 2.2-fold increase in the risk of CKD occurrence, after adjustment for other confounders. The findings suggest that OxLDL may contribute to renal dysfunction, potentially by promoting vascular damage, oxidative stress, inflammation and renin–angiotensin–aldosterone system activation [32]. A case–control study of consecutive patients with AF found that these patients had greater biochemical markers of oxidative stress and inflammation as reflected by OxLDL compared to age- and sex-matched controls [33]. In the current study, AF was not related to changes in lipoprotein distribution in terms of TC, LDL-C, sdLDL-C and OxLDL, whether in whole plasma, the total LDL fraction or sdLDL and lbLDL subclasses.

In the general population, CKD has been linked to varying levels of LDL-C [34], triglycerides, apolipoprotein B and lipoprotein (a) particles, reduced high-density lipoprotein cholesterol (HDL-C) and accumulation of OxLDL [35]. Lipid dysregulation is further implicated among patients with nephrotic syndrome, owing to the degree of proteinuria [36]. Hypertriglyceridaemia occurs in the early stages of CKD, causing an accumulation of VLDL [37], chylomicron remnants and intermediate-density lipoprotein cholesterol [38]. HDL dysfunction and LDL receptor-related protein deficiency increase the chylomicron remnant and intermediate-density lipoprotein cholesterol levels, thereby promoting the formation of sdLDL in patients with CKD [39]. As a result, though serum LDL-C levels may be within the normal range, sdLDL level is thought to increase as kidney function worsens [38]. It was previously reported that dialysis patients had similar TC but lower TG and HDL-C (specifically HDL3-C subtype) and higher OxLDL compared to controls [40]. In the current study, we have demonstrated that among patients with AF, those with CKD had lower TC, LDL-C and sdLDL-C but that these changes were related to the concomitant presence of other comorbidities rather than CKD per se.

### 4.4. Relationship between Fibrin Clot Properties and Lipoprotein Distribution in AF

Lipoproteins and free fatty acids may influence fibrin clot properties via several different mechanisms. Biochemical studies have shown that triglyceride-rich lipoproteins may activate blood coagulation by binding vitamin K-dependent coagulation factors and promoting prothrombinase complex formation, leading to enhanced thrombin generation which renders clots denser and more resistant to lysis [41,42,43]. Additionally, platelet activation results in the release of free fatty acids and lysophospholipid [44], which accumulate within thrombi [45]. The major phospholipid in platelet membranes is lecithin with a high total fatty acid content of oleate and stearate. These free fatty acids have been shown to reduce the mechanical stability of fibrin clot through modulation of thrombin activity and the pattern of fibrin assembly [46]. A study showed that the addition of oxidised HDL reduced clot firmness during thromboelastography using ROTEM analysis with tissue factor, potentially through its effects on platelets rather than fibrin breakdown [47]. Interestingly, the authors reported that no effect on clot firmness was seen with OxLDL.

In a study of patients with primary antiphospholipid syndrome and prior thromboembolism who were matched with controls in terms of age, sex and type of prior thromboembolism, OxLDL was independently related to fibrin permeability [48]. Nonetheless, the study excluded elderly patients and those with AF or prior ischaemic stroke, transient ischaemic attack or coronary events. Among patients with type 2 diabetes, oxidative stress, as reflected by circulating markers including OxLDL, was linked to reduced fibrin permeability and fibrinolysis [49]. However, patients on anticoagulant therapy were excluded from this study. A case–control study of patients with isolated severe aortic stenosis and matched controls found associations between LDL-C and fibrin permeability; triglycerides and clot thickness; and TC, LDL-C, triglycerides and OxLDL and fibrinolysis [50]. Furthermore, eGFR was inversely related to fibrinolytic capacity in this cohort. Nevertheless, patients with AF were excluded from this study [50].

Overall, the effects of lipoproteins on fibrin clot properties warrant further investigation. In particular, evaluation of the relationship of these factors among AF patients with worsening CKD may help identify the prothrombotic mechanisms that contribute to thromboembolic complications and poor clinical outcomes. In the current study, we were unable to attribute the changes in fibrin clot properties in AF patients with worsening CKD to alterations in lipoproteins in terms of TC, LDL-C, sdLDL-C and OxLDL. While the fibrin clot techniques employed in this study are currently not suitable for routine clinical practice, they are helpful in the research setting to help understand the underlying pathophysiological processes in diseases. The significance and interpretation of these biomarkers remains to be determined.

### 4.5. Limitations

There were several limitations to this study that warrant consideration. First, given the observational study design and relatively small sample size, the results should be interpreted with caution. Nonetheless, on the whole, the findings were consistent with other studies. Second, we demonstrate associations related to AF and CKD but not causality. Third, due to the lack of standardisation for turbidimetric clotting analysis and various commercially available ELISA kits for OxLDL analysis, the results may not be generalisable. This highlights the need for normalisation of findings and generation of standardised assays for these techniques to allow direct comparison between laboratories [51]. Fourth, despite adjustment, some residual unmeasured confounders may exist to account for differences observed from this study. Fifth, there were some differences in the choice of anticoagulation agent between patients with AF and the control group, though this did not reach statistical significance.

## 5. Conclusions

In this prospective study on the effects of AF and CKD, there are important changes that occur in fibrin clot properties with both these conditions that may account for the increased risk of thromboembolic complications. However, these changes in fibrin clot properties were not attributable to alterations in lipoprotein distribution, specifically TC, LDL-C, sdLDL-C and OxLDL.

## Figures and Tables

**Table 1 biomedicines-10-02270-t001:** Baseline characteristics of patients with and without atrial fibrillation.

Baseline Characteristics	AF (*n* = 23)	No AF (*n* = 10)	*p* Value
Age (years), median (IQR)	68 (61–76)	74 (51–77)	0.923
Female sex, *n* (%)	9 (39.1%)	2 (20.0%)	0.430
**Smoking status, *n* (%)**			0.083
Current smoker	1 (4.3%)	2 (20.0%)	
Ex-smoker	16 (69.6%)	3 (30.0%)	
Never smoked	6 (26.1%)	5 (50.0%)	
Alcohol intake (units), median (IQR)	2 (0–10)	2 (1–16)	0.524
**Exercise frequency, *n* (%)**			0.642
None	20 (87.0%)	7 (70.0%)	
Weekly	1 (4.3%)	1 (10.0%)	
Daily	2 (8.7%)	2 (20.0%)	
Pulse (bpm), median (IQR)	72 (66–88)	79 (75–85)	0.343
sBP (mmHg), median (IQR)	159 (130–172)	143 (122–163)	0.269
BMI (kg/m^2^), median (IQR)	30.5 (27.6–36.2)	29.8 (25.9–32.3)	0.451
**AF type, *n* (%)**			NA
Paroxysmal	5 (21.7%)	NA	
Persistent	17 (73.9%)	NA	
Permanent	1 (4.3%)	NA	
**Comorbidities, *n* (%)**			
Hypertension	13 (56.5%)	5 (50.0%)	1.000
Hypercholesterolaemia	9 (39.1%)	5 (50.0%)	0.707
Diabetes mellitus	3 (13.0%)	0 (0%)	0.536
Coronary artery disease	5 (21.7%)	1 (10.0%)	0.640
Heart failure	5 (21.7%)	0 (0%)	0.291
Previous stroke or TIA	4 (17.4%)	0 (0%)	0.289
Peripheral artery disease	1 (4.3%)	0 (0%)	1.000
**Anticoagulation, *n* (%)**			0.073
Warfarin	16 (69.6%)	10 (100%)	
NOAC	7 (30.4%)	0 (0%)	
Antiplatelet use, *n* (%)	3 (13.0%)	0 (0%)	0.536
Statin use, *n* (%)	13 (56.5%)	5 (50.0%)	1.000
**Biochemical Parameters**			
**Full blood count**			
Haemoglobin (g/L)	143 (139–156)	135 (125–149)	0.287
White cell count (10^9^ per L)	6.5 (6.3–8.0)	6.9 (6.0–8.0)	0.893
Platelet count (10^9^ per L)	219 (165–250)	221 (164–274)	0.954
MCV (fL)	91.4 (88.2–94.9)	93.4 (87.4–96.5)	0.743
Haematocrit (%)	42.3 (40.2–46.1)	40.8 (38.9–45.6)	0.428
Neutrophils (10^9^ per L)	4.2 (3.2–6.2)	4.0 (3.5–4.8)	0.475
Lymphocytes (10^9^ per L)	1.6 (1.1–2.1)	1.9 (1.9–2.4)	0.051
**Renal profile**			
Sodium (mmol/L)	140 (139–141)	140 (138–141)	0.499
Potassium (mmol/L)	4.2 (4.1–4.5)	4.4 (4.1–4.7)	0.524
Urea (mmol/L)	6.3 (4.5–8.1)	4.4 (3.3–6.9)	0.133
Creatinine (µmol/L)	80 (69–104)	86 (69–95)	0.862
eGFR (ml/min/1.73 m^2^)	72 (59–91)	78 (66–101)	0.343
**Liver function tests**			
Bilirubin (µmol/L)	10 (7–15)	9 (7–13)	0.406
Alkaline phosphate (U/L)	83 (64–116)	79 (63–85)	0.451
ALT (U/L)	23 (19–30)	18 (17–24)	0.105
Albumin (g/L)	43 (41–46)	43 (41–44)	0.603
Clotting profile			
INR	2.3 (1.7–2.6)	2.4 (2.0–2.9)	0.343
Fibrinogen (g/L)	3.4 (3.1–4.3)	3.3 (3.1–3.7)	0.345
CRP (mg/L)	5 (5–8)	5 (2–5)	0.363
Calcium (mmol/L)	2.4 (2.3–2.5)	2.4 (2.3–2.5)	0.923

AF, atrial fibrillation; ALT, alanine transaminase; BMI, body mass index; CRP, C-reactive protein; eGFR, estimated glomerular filtration rate; INR, international normalised ratio; IQR, interquartile range; MCV, mean corpuscular volume; NA, not applicable; NOAC, non-vitamin K antagonist oral anticoagulant; sBP, systolic blood pressure; TIA, transient ischaemic attack.

**Table 2 biomedicines-10-02270-t002:** Baseline characteristics of AF patients with and without chronic kidney disease.

Baseline Characteristics	CKD (*n* = 23)	No CKD (*n* = 23)	*p* Value
Age (years), median (IQR)	78 (69–84)	68 (61–76)	0.012
Female sex, *n* (%)	8 (34.8%)	9 (39.1%)	0.760
**Smoking status, *n* (%)**			0.597
Current smoker	0 (0%)	1 (4.3%)	
Ex-smoker	17 (73.9%)	16 (69.6%)	
Never smoked	6 (26.1%)	6 (26.1%)	
Alcohol intake (units), median (IQR)	2 (0–4)	2 (0–10)	0.337
**Exercise frequency, *n* (%)**			0.639
None	20 (87.0%)	20 (87.0%)	
Weekly	2 (8.7%)	1 (4.3%)	
Daily	1 (4.3%)	2 (8.7%)	
Pulse (bpm), median (IQR)	79 (68–85)	72 (66–88)	0.676
sBP (mmHg), median (IQR)	150 (132–173)	159 (130–172)	0.939
BMI (kg/m^2^), median (IQR)	29.7 (24.0–36.0)	30.5 (27.6–36.2)	0.307
**AF type, *n* (%)**			0.151
Paroxysmal	1 (4.3%)	5 (21.7%)	
Persistent	19 (82.6%)	17 (73.9%)	
Permanent	3 (13.0%)	1 (4.3%)	
**Comorbidities, *n* (%)**			
Hypertension	18 (78.3%)	13 (56.5%)	0.116
Hypercholesterolaemia	11 (47.8%)	9 (39.1%)	0.552
Diabetes mellitus	12 (52.2%)	3 (13.0%)	0.005
Coronary artery disease	5 (21.7%)	5 (21.7%)	1.000
Heart failure	13 (56.5%)	5 (21.7%)	0.016
Previous stroke or TIA	4 (17.4%)	3 (13.0%)	1.000
Peripheral artery disease	1 (4.3%)	1 (4.3%)	1.000
**Anticoagulation agent, *n* (%)**			0.743
Warfarin	17 (73.9%)	16 (69.6%)	
NOAC	6 (26.1%)	7 (30.4%)	
Antiplatelet use, *n* (%)	1 (4.3%)	3 (13.0%)	0.608
Statin use, *n* (%)	20 (87.0%)	13 (56.5%)	0.022
**Biochemical Parameters**			
**Full blood count**			
Haemoglobin (g/L)	130 (119–143)	143 (139–156)	0.017
White cell count (10^9^ per L)	8.1 (6.8–9.7)	6.5 (6.3–8.0)	0.080
Platelet count (10^9^ per L)	216 (181–246)	219 (165–250)	0.965
MCV (fL)	92.3 (89.8–98.4)	91.4 (88.2–94.9)	0.429
Haematocrit (%)	40.0 (35.6–43.7)	42.3 (40.2–46.1)	0.037
Neutrophils (10^9^ per L)	5.1 (4.4–6.6)	4.2 (3.2–6.2)	0.084
Lymphocytes (10^9^ per L)	1.8 (1.3–2.4)	1.6 (1.1–2.1)	0.545
**Renal profile**			
Sodium (mmol/L)	141 (139–143)	140 (139–141)	0.417
Potassium (mmol/L)	4.3 (4.0–4.8)	4.2 (4.1–4.5)	0.440
Urea (mmol/L)	11.6 (9.3–12.6)	6.3 (4.5–8.1)	<0.001
Creatinine (µmol/L)	164 (121–200)	80 (69–104)	<0.001
eGFR (ml/min/1.73 m^2^)	32 (24–40)	72 (59–91)	<0.001
**Liver function tests**			
Bilirubin (µmol/L)	10 (7–14)	10 (7–15)	0.891
Alkaline phosphate (U/L)	80 (63–122)	83 (64–116)	0.750
ALT (U/L)	16 (14–22)	23 (19–30)	0.003
Albumin (g/L)	44 (41–46)	43 (41–46)	0.921
**Clotting profile**			
INR	2.0 (1.2–2.2)	2.3 (1.7–2.6)	0.129
Fibrinogen (g/L)	3.6 (3.1–4.6)	3.4 (3.1–4.3)	0.262
CRP (mg/L)	5 (2–7)	5 (5–8)	0.847
Calcium (mmol/L)	2.4 (2.3–2.5)	2.4 (2.3–2.5)	0.489

AF, atrial fibrillation; ALT, alanine transaminase; BMI, body mass index; CRP, C-reactive protein; eGFR, estimated glomerular filtration rate; INR, international normalised ratio; IQR, interquartile range; MCV, mean corpuscular volume; NA, not applicable; NOAC, non-vitamin K antagonist oral anticoagulant; sBP, systolic blood pressure; TIA, transient ischaemic attack.

**Table 3 biomedicines-10-02270-t003:** Fibrin clot properties among patients with and without atrial fibrillation.

Turbidimetric Clotting Analysis	AF (*n* = 23)	No AF (*n* = 10)	*p* Value
**Kaolin reagent**			
T_lag_ (s)	1640 (1154–1986)	1794 (630–1970)	0.883
Abs_peak_ (units, 340 nm)	0.084 (0.066–0.142)	0.091 (0.036–0.108)	0.476
T_max_ (s)	1962 (1646–2418)	2042 (1219–2306)	0.836
T_peak50%_ (s)	2046 (1526–2314)	2004 (1334–2294)	0.856
T_lysis50%_ (s)	368 (170–834)	280 (260–492)	0.711
**PPP reagent**			
T_lag_ (s)	748 (586–934)	626 (534–694)	0.411
Abs_peak_ (units, 340 nm)	0.141 (0.104–0.215)	0.116 (0.103–0.146)	0.180
T_max_ (s)	1325 (918–1502)	1008 (814–1482)	0.589
T_peak50%_ (s)	998 (690–1510)	824 (626–1130)	0.617
T_lysis50%_ (s)	904 (308–1506)	220 (128–378)	0.070
**PTT reagent**			
T_lag_ (s)	314 (304–354)	358 (346–366)	0.047
Abs_peak_ (units, 340 nm)	0.153 (0.102–0.240)	0.111 (0.097–0.126)	0.031
T_max_ (s)	618 (532–714)	556 (506–630)	0.363
T_peak50%_ (s)	394 (376–446)	432 (402–454)	0.114
T_lysis50%_ (s)	884 (276–1556)	280 (144–496)	0.047
**Thrombin reagent**			
T_lag_ (s)	170 (138–486)	132 (126–146)	0.031
Abs_peak_ (units, 340 nm)	0.109 (0.088–0.134)	0.101 (0.080–0.131)	0.630
T_max_ (s)	590 (470–894)	462 (442–518)	0.047
T_peak50%_ (s)	406 (218–970)	220 (198–254)	0.005
T_lysis50%_ (s)	348 (140–616)	154 (84–460)	0.483
Ks (10^−9^ cm^2^)	6.33 (3.37–12.20)	8.84 (5.92–11.19)	0.633

Abs_peak_, peak absorbance; AF, atrial fibrillation; Ks, permeation constant; T_lag_, lag time; T_lysis50%_, time to 50% clot lysis; T_max_, time to maximum absorbance; T_peak50%_, time to 50% peak absorbance.

**Table 4 biomedicines-10-02270-t004:** Fibrin clot properties among atrial fibrillation patients with and without chronic kidney disease.

Turbidimetric Clotting Analysis	CKD (*n* = 23)	No CKD (*n* = 23)	*p* Value
**Kaolin reagent**			
T_lag_ (s)	1072 (688–1250)	1640 (1154–1986)	0.003
Abs_peak_ (units, 340 nm)	0.112 (0.019–0.152)	0.084 (0.066–0.142)	0.923
T_max_ (s)	1458 (998–1734)	1962 (1646–2418)	0.005
T_peak50%_ (s)	1294 (842–1806)	2046 (1526–2314)	0.008
T_lysis50%_ (s)	460 (224–536)	368 (170–834)	0.948
**PPP reagent**			
T_lag_ (s)	566 (510–646)	748 (586–934)	0.044
Abs_peak_ (units, 340 nm)	0.143 (0.119–0.184)	0.141 (0.104–0.215)	0.701
T_max_ (s)	922 (826–1210)	1325 (918–1502)	0.134
T_peak50%_ (s)	702 (638–898)	998 (690–1510)	0.073
T_lysis50%_ (s)	512 (356–756)	904 (308–1506)	0.318
**PTT reagent**			
T_lag_ (s)	302 (290–366)	314 (304–354)	0.252
Abs_peak_ (units, 340 nm)	0.141 (0.116–0.206)	0.153 (0.102–0.240)	0.852
T_max_ (s)	546 (494–610)	618 (532–714)	0.124
T_peak50%_ (s)	394 (358–454)	394 (376–446)	0.676
T_lysis50%_ (s)	900 (396–1324)	884 (276–1556)	0.517
**Thrombin reagent**			
T_lag_ (s)	290 (154–378)	170 (138–486)	0.302
Abs_peak_ (units, 340 nm)	0.117 (0.091–0.182)	0.109 (0.088–0.134)	0.482
T_max_ (s)	694 (510–1206)	590 (470–894)	0.156
T_peak50%_ (s)	622 (254–706)	406 (218–970)	0.489
T_lysis50%_ (s)	316 (192–828)	348 (140–616)	0.427
Ks (10^−9^ cm^2^)	5.81 (3.55–9.01)	6.33 (3.37–12.20)	0.297

Abs_peak_, peak absorbance; CKD, chronic kidney disease; Ks, permeation constant; T_lag_, lag time; T_lysis50%_, time to 50% clot lysis; T_max_, time to maximum absorbance; T_peak50%_, time to 50% peak absorbance.

**Table 5 biomedicines-10-02270-t005:** Lipoprotein distribution of patients with and without atrial fibrillation.

Lipoprotein Distribution	AF (*n* = 23)	No AF (*n* = 10)	*p* Value
**Whole plasma**			
Total cholesterol (mmol/L)	4.63 (3.99–5.47)	4.63 (3.66–5.33)	0.576
LDL-C (mmol/L)	2.97 (2.14–3.72)	2.35 (1.73–3.36)	0.324
sdLDL (mmol/L)	0.64 (0.41–0.96)	0.63 (0.30–0.94)	0.630
OxLDL (ng/mL)	71.28 (54.76–88.92)	89.04 (61.85–124.92)	0.180
**Total LDL**			
Total cholesterol (mmol/L)	5.60 (4.34–6.11)	4.43 (3.45–5.27)	0.269
LDL-C (mmol/L)	4.57 (3.50–5.23)	3.38 (2.71–3.83)	0.253
sdLDL (mmol/L)	0.60 (0.47–0.97)	0.61 (0.47–1.20)	0.923
OxLDL (ng/mL)	18.03 (10.24–32.99)	20.30 (15.94–20.84)	0.954
**lbLDL subclass**			
Total cholesterol (mmol/L)	6.49 (4.76–8.26)	5.41 (4.29–6.67)	0.428
LDL-C (mmol/L)	5.83 (2.99–7.61)	4.06 (3.55–5.18)	0.363
sdLDL (mmol/L)	0.60 (0.47–0.70)	0.53 (0.47–0.83)	0.832
OxLDL (ng/mL)	26.18 (12.34–46.06)	26.69 (25.18–34.87)	0.923
**sdLDL subclass**			
Total cholesterol (mmol/L)	2.99 (2.01–4.11)	2.85 (1.54–4.43)	0.893
LDL-C (mmol/L)	2.29 (1.87–3.59)	2.29 (1.87–3.64)	0.893
sdLDL (mmol/L)	0.71 (0.47–1.13)	0.86 (0.47–1.77)	0.686
OxLDL (ng/mL)	11.88 (4.74–28.98)	11.55 (9.21–14.31)	0.954

AF, atrial fibrillation; lbLDL, large buoyant low-density lipoprotein; LDL-C, low-density lipoprotein cholesterol; OxLDL, oxidised low-density lipoprotein; sdLDL, small dense low-density lipoprotein.

**Table 6 biomedicines-10-02270-t006:** Lipoprotein distribution of atrial fibrillation patients with and without chronic kidney disease.

Lipoprotein Distribution	CKD (*n* = 23)	No CKD (*n* = 23)	*p* Value
**Whole plasma**			
Total cholesterol (mmol/L)	4.16 (3.34–4.82)	4.63 (3.99–5.47)	0.022
LDL-C (mmol/L)	2.03 (1.37–2.66)	2.97 (2.14–3.72)	0.002
sdLDL (mmol/L)	0.43 (0.23–0.75)	0.64 (0.41–0.96)	0.016
OxLDL (ng/mL)	69.57 (51.79–95.16)	71.28 (54.76–88.92)	0.956
**Total LDL**			
Total cholesterol (mmol/L)	3.31 (2.80–4.95)	5.60 (4.34–6.11)	0.002
LDL-C (mmol/L)	2.57 (2.01–4.06)	4.57 (3.50–5.23)	0.003
sdLDL (mmol/L)	0.47 (0.47–0.67)	0.60 (0.47–0.97)	0.031
OxLDL (ng/mL)	24.28 (18.15–35.43)	18.03 (10.24–32.99)	0.362
**lbLDL subclass**			
Total cholesterol (mmol/L)	4.01 (3.45–5.18)	6.49 (4.76–8.26)	0.001
LDL-C (mmol/L)	2.94 (2.43–3.45)	5.83 (2.99–7.61)	0.002
sdLDL (mmol/L)	0.47 (0.47–0.61)	0.60 (0.47–0.70)	0.021
OxLDL (ng/mL)	31.87 (21.27–40.87)	26.18 (12.34–46.06)	0.560
**sdLDL subclass**			
Total cholesterol (mmol/L)	1.96 (1.40–3.03)	2.99 (2.01–4.11)	0.028
LDL-C (mmol/L)	1.87 (1.87–2.61)	2.29 (1.87–3.59)	0.070
sdLDL (mmol/L)	0.47 (0.47–0.73)	0.71 (0.47–1.13)	0.041
OxLDL (ng/mL)	16.20 (10.05–31.45)	11.88 (4.74–28.98)	0.475

lbLDL, large buoyant low-density lipoprotein; LDL-C, low-density lipoprotein cholesterol; OxLDL, oxidised low-density lipoprotein; sdLDL, small dense low-density lipoprotein.

**Table 7 biomedicines-10-02270-t007:** Correlation between fibrin clot properties and lipoprotein distribution in whole plasma of patients with atrial fibrillation.

	Pearson’s Correlation Coefficient (95% CI)			
	Cholesterol	LDL-C	sdLDL-C	OxLDL
**Kaolin reagent**				
T_lag_	−0.072 (−0.400 to 0.273)	0.057 (−0.286 to 0.388)	0.049 (−0.294 to 0.381)	0.167 (−0.181 to 0.478)
Abs_peak_	−0.102 (−0.391 to 0.204)	−0.189 (−0.463 to 0.118)	−0.263 (−0.522 to 0.041)	−0.061 (−0.355 to 0.243)
T_max_	−0.096 (−0.420 to 0.250)	0.001 (−0.337 to 0.339)	−0.007 (−0.345 to 0.332)	0.102 (−0.245 to 0.425)
T_peak50%_	−0.023 (−0.340 to 0.299)	0.126 (−0.202 to 0.428)	0.136 (−0.192 to 0.436)	0.116 (−0.212 to 0.420)
T_lysis50%_	0.128 (−0.251 to 0.472)	0.243 (−0.135 to 0.560)	0.199 (−0.181 to 0.527)	−0.002 (−0.368 to 0.365)
**PPP reagent**				
T_lag_	0.115 (−0.185 to 0.395)	0.167 (−0.133 to 0.439)	0.312 (0.020 to 0.555)	−0.001 (−0.295 to 0.292)
Abs_peak_	−0.207 (−0.469 to 0.089)	−0.127 (−0.403 to 0.169)	−0.206 (−0.468 to 0.090)	−0.116 (−0.393 to 0.180)
T_max_	−0.020 (−0.312 to 0.275)	0.065 (−0.233 to 0.352)	0.153 (−0.147 to 0.428)	−0.040 (−0.330 to 0.256)
T_peak50%_	0.066 (−0.232 to 0.352)	0.142 (−0.158 to 0.418)	0.232 (−0.066 to 0.492)	−0.069 (−0.355 to 0.230)
T_lysis50%_	0.138 (−0.162 to 0.415)	0.178 (−0.122 to 0.448)	0.178 (−0.122 to 0.448)	−0.136 (−0.413 to 0.165)
**PTT reagent**				
T_lag_	0.046 (−0.248 to 0.332)	0.053 (−0.241 to 0.338)	−0.065 (−0.349 to 0.229)	0.359 (0.077 to 0.588)
Abs_peak_	−0.234 (−0.491 to 0.060)	−0.138 (−0.412 to 0.159)	−0.169 (−0.438 to 0.128)	−0.051 (−0.337 to 0.243)
T_max_	−0.069 (−0.352 to 0.226)	−0.011 (−0.300 to 0.280)	−0.127 (−0.402 to 0.170)	0.207 (−0.089 to 0.469)
T_peak50%_	0.037 (−0.256 to 0.324)	0.079 (−0.217 to 0.361)	−0.003 (−0.293 to 0.287)	0.287 (−0.004 to 0.533)
T_lysis50%_	0.140 (−0.157 to 0.413)	0.136 (−0.160 to 0.410)	0.221 (−0.074 to 0.481)	−0.140 (−0.413 to 0.157)
**Thrombin reagent**				
T_lag_	0.008 (−0.283 to 0.298)	0.004 (−0.286 to 0.294)	−0.066 (−0.350 to 0.228)	0.056 (−0.238 to 0.341)
Abs_peak_	−0.150 (−0.422 to 0.147)	−0.121 (−0.397 to 0.175)	−0.197 (−0.461 to 0.099)	−0.029 (−0.317 to 0.263)
T_max_	−0.195 (−0.459 to 0.101)	−0.156 (−0.427 to 0.141)	−0.232 (−0.489 to 0.063)	0.052 (−0.242 to 0.337)
T_peak50%_	−0.044 (−0.330 to 0.250)	−0.044 (−0.330 to 0.250)	−0.170 (−0.438 to 0.127)	0.063 (−0.231 to 0.347)
T_lysis50%_	−0.024 (−0.316 to 0.271)	−0.062 (−0.350 to 0.235)	−0.058 (−0.346 to 0.240)	0.049 (−0.248 to 0.337)
Ks	0.090 (−0.224 to 0.387)	0.125 (−0.190 to 0.416)	−0.043 (−0.346 to 0.268)	0.168 (−0.148 to 0.452)

Abs_peak_, peak absorbance; Ks, permeation constant; T_lag_, lag time; T_lysis50%_, time to 50% clot lysis; T_max_, time to maximum absorbance; tPA, tissue plasminogen activator; T_peak50%_, time to 50% peak absorbance.

## Data Availability

The data underlying this article will be shared on reasonable request to the corresponding author.

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
