# Peer review of "Relationship between Renal Function, Fibrin Clot Properties and Lipoproteins in Anticoagulated Patients with Atrial Fibrillation"

_biomedicines, 2022, doi:10.3390/biomedicines10092270_

Round 1

Reviewer 1 Report

In the paper the authors describe relationship between atrial fibrillation, renal function and fibrin clot properties in anticoagulated patients.

Total population included 56 patients. The first group included 46 patients with atrial fibrillation (AF), 23 with chronic kidney disease (CKD) and 23 without CKD, all the patients was anticoagulated. The control group included 10 anticoagulated patients without AF or CKD.

Overall, authors should define more accurately the potential clinical impact of their research (would they suggest to measure these biomarkers in all the patients? would the clinical management differ according to the results of their study?) Would the authors think that it is a cost effective analysis?

I have the following comments:

·       How were the patients enrolled in the prospective cross-sectional study? Consecutively, randomly, other? Please specify

·       In the baseline characteristics the authors should include also the presence of stroke/transient ischemic attack and peripheral artery disease that, as diabetes, heart failure and coronary artery disease are very important findings influencing the pro thrombotic status

·       The authors may describe that the anticoagulation therapy substantially differs between the study group and the control group because in the first about 30% of patients were on non-vitamin K antagonist oral anticoagulants, whether in the control group all the patients were on vitamin K antagonist oral anticoagulant therapy.

·       The authors should report the indication for anticoagulation in control group (pulmonary embolism? Deep venous thrombosis? Other?).

·       What about definition of AF patients? Were patients with paroxysmal AF, permanent AF, persistent AF? Please specify

·       In the text was reported “This was the first study to demonstrate serial changes in fibrin clot structure in AF across worsening degrees of renal function”, but the relative table is reported as supplementary materials. In fact, in the Supplementary Table 3 are reported the results of correlation analyses between eGFR and fibrin clot properties in AF; the most important findings could be reported as graphic in the main text and the rest as Supplementary Table (as it is)

·       Moreover, in the manuscript there was not indicated the degrees of renal function but only their impairment indicated as eGFR < 50 ml/min an there is not a stratification of degree of renal function. I would therefore suggest to evaluate the difference among patients with the different degrees of renal function

·     If allowed, the inclusion of a graphical abstract or main image would give the article greater visual impact.

·       Finally, it should be indicated in the title that the study population consists of patients on anticoagulation therapy

Reviewer 2 Report

As commented by the authors in Limitations, it is about an obsevational study with small sample size and only may be described associations, but not causality. For this reason "investigate the effects of AF and CKD on fibrin clot properties and lipoproteins......" is not appropriate enough to describe the objectives of the study

Round 2

Reviewer 1 Report

Some of the answers are very poorly detailed. The classification of atrial fibrillation and the presentation of other clinical features that may influence the results are lacking in detail. 

I would suggest to reconsider the title because clinical features of atrial fibrillation are not clear or, alternatively, deepening these in the text (Paroxysmal? Persistent? Permanent? Atrial fibrillation associated or not with structural heart disease?). The work is very complete as far as the laboratory part is concerned, probably the title should mainly bring out this feature.

Round 3

Reviewer 1 Report

The authors replied to the comments

Author Response

thanks